# Evaluation of the BD Phoenix Carbapenemase-Producing Organism Panels for the Detection of Carbapenemase Producers in *Escherichia coli, Klebsiella pneumoniae* and *Pseudomonas aeruginosa*

**DOI:** 10.3390/diagnostics13223417

**Published:** 2023-11-09

**Authors:** Yoselin Paola Correa-León, José Miguel Pérez-Hernández, Bernardo Alfonso Martinez-Guerra, Eduardo Rodríguez-Noriega, Juan Pablo Mena-Ramírez, Eduardo López-Gutiérrez, Luis Esaú López-Jácome, Víctor Antonio Monroy-Colin, Christian Daniel Mireles-Davalos, Cecilia Padilla-Ibarra, María Angelina Quevedo-Ramos, José Manuel Feliciano-Guzmán, Talía Pérez-Vicelis, María del Consuelo Velázquez-Acosta, Melissa Hernández-Durán, Elvira Garza-González

**Affiliations:** 1Departamento de Bioquímica, Facultad de Medicina, Universidad Autónoma de Nuevo León, Monterrey 66460, Mexico; paola.correa2310@gmail.com (Y.P.C.-L.); jmph141298@gmail.com (J.M.P.-H.); 2Instituto Nacional de Ciencias Médicas y Nutrición Salvador Zubirán, Mexico City 14080, Mexico; beramg@gmail.com; 3Hospital Civil de Guadalajara, Instituto de Patología Infecciosa y Experimental “Dr. Francisco Ruiz Sánchez”, Centro Universitario Ciencias de la Salud, Universidad de Guadalajara, Guadalajara 44280, Mexico; idfcolima@yahoo.com; 4Laboratorio de Microbiología, Hospital General de Zona No. 21 IMSS Tepatitlán de Morelos, Jalisco, Centro Universitario de los Altos (Cualtos), Universidad de Guadalajara, Guadalajara 47630, Mexico; dr.juanmena@gmail.com; 5Laboratorio de Microbiología, Hospital Regional de Alta Especialidad de Oaxaca, Oaxaca de Juárez 71256, Mexico; campos.5488@gmail.com; 6Laboratorio de Infectología, Instituto Nacional de Rehabilitación Luis Guillermo Ibarra Ibarra, Mexico City 14389, Mexicomelypsp@yahoo.com.mx (M.H.-D.); 7Laboratorio de Microbiología, Centenario Hospital Miguel Hidalgo, Aguascalientes 20259, Mexico; vmonroyc@yahoo.com.mx; 8Laboratorio de Microbiología, Instituto Nacional de Enfermedades Respiratorias Ismael Cosío Villegas, Mexico City 14080, Mexico; cdmirelesd@hotmail.com; 9Laboratorio de Microbiología, Hospital General del Estado de Sonora, Hermosillo 83249, Mexico; ceciliapadillaibarra@gmail.com; 10Laboratorio de Microbiología, Hospital General de León, León 37672, Mexico; maquevedor@hotmail.com; 11Laboratorio de Microbiología, Hospital de Especialidades Pediátricas, Tuxtla Gutiérrez 29070, Mexico; jmfelguz12@gmail.com; 12Laboratorio de Microbiología, Hospital Regional de Alta Especialidad Bicentenario de la Independencia, Tultitlan 54916, Mexico; ailat.perevic@gmail.com; 13Laboratorio de Microbiología, Instituto Nacional de Cancerología, Mexico City 14080, Mexico; consueve62@yahoo.com.mx

**Keywords:** carbapenemase producer organisms, NDM, KPC, CO panels

## Abstract

The classification of carbapenemases can help guide therapy. The present study evaluated the performance of the CPO detection test, included in the BD Phoenix^™^ NMIC-501 panel for the detection and classification of carbapenemases on the representative molecularly characterized strains collection from Mexico. Carbapenem non-susceptible isolates collected in Mexico were included. The clinical isolates (*n* = 484) comprised *Klebsiella pneumoniae* (*n* = 154), *Escherichia coli* (*n* = 150), and *P. aeruginosa* (*n* = 180). BD Phoenix CPO NMIC-504 and NMIC-501 panels were used for the identification of species, antimicrobial susceptibility tests, and detection of CPOs. For the detection of carbapenemase-encoding genes, *E. coli* and *K. pneumoniae* were evaluated using PCR assays for *bla*_NDM-1_, *bla*_KPC_, *bla*_VIM_, *bla*_IMP_, and *bla*_OXA-48-like_. For *P. aeruginosa*, *bla*VIM, *bla*IMP, and *bla*GES were detected using PCR. Regarding *E. coli*, the CPO panels had a sensitivity of 70% and specificity of 83.33% for the detection of a class B carbapenemase (*bla*_NDM_ in the molecular test). Regarding *K. pneumoniae*, the panels had a sensitivity of 75% and specificity of 100% for the detection of a class A carbapenemase (*bla*_KPC_ in the molecular test). The Phoenix NMIC-501 panels are reliable for detecting class B carbapenemases in *E. coli*. The carbapenemase classification in *K. pneumoniae* for class A carbapenemases has a high specificity and PPV; thus, a positive result is of high value.

## 1. Introduction

Reports on the spread of carbapenem-resistant Gram-negative pathogens, including carbapenem-resistant Enterobacterales (CRE) and some non-fermenters such as *Pseudomonas aeruginosa* [1], are increasing in many countries. Bacteria can develop carbapenem resistance to antibiotics via some mechanisms, including the presence of porins and efflux pumps, the modification of the target of the antibiotics, and the action of enzymes, especially β-lactamases, which can inactivate carbapenems. The production of carbapenemases is of significant concern because these enzymes are often encoded on mobile genetic elements, including plasmids, prophages, pathogenicity islands, transposons, and others, and, in some cases, can spread rapidly via lateral gene transfer [1,2,3].

β-lactamases are grouped into classes A, B, C, and D according to the Ambler classification. Class A, C, and D enzymes have serine as an enzyme active center, whereas class B enzymes use the metal zinc [2]. Among class A β-lactamases, *Klebsiella pneumoniae* carbapenemase (KPC) hydrolyzes penicillins, cephalosporins, and carbapenems and is inhibited by clavulanate or tazobactam [3]. Class B β-lactamases (*bla*IMP, *bla*NDM, and *bla*VIM are the most prevalent) have activity against all beta-lactams except aztreonam and are not inhibited by clavulanate or tazobactam. Group C includes cephalosporinases, and Group D is oxacillinases, including OXA-48, OXA-23, and OXA-24 [3].

The classification of carbapenemases has been helpful in guiding therapy since 2015. There are new β-lactamase inhibitor combinations that include avibactam, relebactam, and vaborbactam that have a spectrum limited to class A and, in some cases, class D carbapenemases. Only monobactams, such as aztreonam, are stable to class B metallo-β-lactamases [4].

The worldwide distribution of carbapenemase-producing organisms (CPOs) has led to the development of diagnostic tools to obtain results in a shorter period of time. In bacteriology laboratories, based on the results of antibiotic susceptibility testing (AST), suspicion of the presence of CPO may arise, and phenotypic or molecular-based methods can be used to study the presence of carbapenemases. The most used phenotypic tests include the Carba NP test [5], the carbapenem inactivation method [6], and lateral flow immunoassay tests [7,8].

Phoenix AST panels^®^ (Becton-Dickinson [BD], Sparks, NV, USA) have been widely used for many years, with some panels available in different regions [9,10]. Recently, the Phoenix CPO Detect Test panels for the identification and classification of class A, B, and D carbapenemases, along with simultaneous AST for Gram-negative bacteria, were developed [1,11], and some studies have investigated its performance [11,12,13,14,15,16,17]. The CPO panels include several carbapenems and other antibiotics previously reported to have activity on some Ambler classes, including temocillin and cloxacillin. Furthermore, these panels include some β-lactamase inhibitors to characterize the carbapenemases according to the Ambler classification [18]. The present study evaluated the performance of the Phoenix CPO detection test included in the BD Phoenix™ NMIC-501 panel for the detection and classification of carbapenemases on a collection of previously characterized strains collected from Mexico.

## 2. Materials and Methods

### 2.1. Bacterial Strains

Clinical isolates with non-susceptibility to any of the carbapenems included in panels were studied. The isolates were collected from clinical samples between January 2021 and January 2023 from 12 centers in Mexico. The clinical isolates (*n* = 484) comprised species of Gram negatives including *Klebsiella pneumoniae* (*n* = 154), *Escherichia coli* (*n* = 150), and *P. aeruginosa* (*n* = 180). Strains were stored at −80 °C until they were used. Bacteria were cultured in MacConkey agar, blood agar, and CHROMagar (BD) at 35 ± 2 °C for 18–24 h.

### 2.2. Assay Using Phoenix CPO Panels

BD Phoenix CPO NMIC-504 panels were used for the genus and species identification and NMIC-501 panels for AST and carbapenemase detection. The NMIC-501 panels provide the Ambler classification for Enterobacterales and non-fermenting organisms. Assays using the panels were performed according to the manufacturer’s instructions. Briefly, colonies were suspended in a BD Phoenix ID broth, and the suspension was adjusted to a McFarland standard turbidity index of approximately 0.25. Automated inoculum preparation and standardization were performed using the BD Phoenix^™^ AP instrument combined with the BD Phoenix^™^ M50 instrument, following the manufacturer’s recommendations.

The resulting broth was then added to the AST panel, which was then loaded into the Phoenix instrument. The results were analyzed using the EpiCenter data management software package (v. 6.61A; BD Diagnostic Systems). The system automatically performed assays, determined the minimum inhibitory concentrations, detected CPOs, and provided Ambler classification results based on the specifications of each panel. Carbapenem Non-susceptibility was determined according to the Clinical and Laboratory Standards Institute [19].

### 2.3. Molecular Characterization of Carbapenemases in Gram-Negatives

For the detection of carbapenemase-encoding genes, a DNA template was prepared, resuspending two colonies in distilled water (50 μL). After, the suspension was heated at 95 °C for 10 min. Tubes were centrifuged, and the supernatant was diluted 1:10 in distilled water and used for detection of genes. Isolates were evaluated using PCR assays in *E. coli* and *K. pneumoniae* for *bla*_NDM-1_, *bla*_KPC_, *bla*_VIM_, *bla*_IMP_, and *bla*_OXA-48-like_ and in *P. aeruginosa* for *bla*_VIM,_ *bla*_IMP_, and *bla*_GES_, as described in previous studies [20,21] (Appendix A). All PCR products were detected using agarose gel electrophoresis. For positive controls, we used strains ATCC BAA2468 and ATCC BAA1905 and clinical isolates previously characterized as P104, LMM-1873, LMM 1105, LMM 2935.

### 2.4. Diagnostic Utility

The sensitivity, specificity, positive likelihood ratio (PLR), negative likelihood ratio (NLR), prevalence, positive predictive value (PPV), negative predictive value (NPV), and accuracy were evaluated by comparing the CPO results with the presence of carbapenemase-encoding genes. The methodology is described in Figure 1.

## 3. Results

### 3.1. Detected Carbapenemase-Encoding Genes

For all three bacterial species, the carbapenemases reported were class A, B, and D carbapenemase producers. Some reports had no recommendations; thus, there was no indication of the presence of carbapenemase in the report of the Phoenix instrument (Table 1, Table 2 and Table 3).

Regarding *E. coli*, the most frequent carbapenemases reported in the CPO panel were class B carbapenemases (*n* = 89/150) (Table 1). Among them, 84 tested positive for *bla*_NDM,_ and 1 was *bla*_VIM_. Furthermore, four strains were found to be positive for *bla*_NDM_ with no recommendation or positive result for class A or D. Furthermore, 34 strains had no recommendation in the CPO panels and were positive for *bla*_NDM._

Regarding *K. pneumoniae*, 18/154 strains had a class A carbapenemase report (Table 2), and *bla*_KPC_ was detected in all of them. Moreover, in 72/154 strains, class B was detected in the CPO panels; among them, *bla*_NDM_ was detected in 65, while *bla*_VIM_ was detected in 1. There was no recommendation in 60/154 strains: 11 had no carbapenemase-encoding gene detected, while 49 had at least one such gene detected.

Regarding *P. aeruginosa*, class A was reported in the CPO panels for 13/180 strains (15 were positive for *bla*_GES_, and 6 had no gene detected) (Table 3). Furthermore, class B was detected in the CPO panels for 38/180 strains, while no class B carbapenemase-encoding gene was detected for 6 strains. Furthermore, 18 strains were reported as carbapenemase producers in the CPO panels, with 6 having no positive result in the PCR assays. Finally, 52 strains had no recommendation, with 39 being negative for the genes studied and 13 having at least one gene detected.

### 3.2. Performance of NMIC-501 CPO Panels

Regarding *E. coli*, the CPO panels had a sensitivity of 70% and specificity of 83.33% for the detection of a class B carbapenemase (*bla*_NDM_ in the molecular test). The PPV value was 94.38, and the NPV was 40.98. Regarding *K. pneumoniae*, the panels had a sensitivity of 75% and specificity of 100% for the detection of a class A carbapenemase (*bla*_KPC_ in the molecular test). The PPV value was 100.00, and the NPV was 95.59. For *E. coli* and *K. pneumoniae* combined, the panels had a sensitivity of 63.14%, and their specificity was 82.35% for detecting class B carbapenemases. Lastly, regarding *P. aeruginosa*, the panels had a sensitivity of 23.35% and specificity of 74.51% for no carbapenemase recommendation (Table 4).

According to our results, the highest positive likelihood ratio observed was for the detection of carbapenemase class B in *E. coli*, which was 4.20 (95% CI, 1.87–9.43), and the lowest was for no carbapenemase recommendation in *P. aeruginosa (*0.92, 95% CI 0.53–1.58).

Regarding the negative likelihood ratio, the lowest was for the detection of a class A carbapenemase in *K. pneumoniae* (0.25, 95% CI 0.13–0.50), and the highest was for no carbapenemase recommendation in *P. aeruginosa (*1.03, 95% CI 0.86–1.23)

In this evaluation, the best accuracy was observed for the detection of a class A carbapenemase in *K. pneumoniae (*96.10, 95% CI 91.71–98.56), and, as expected, the lowest value was for no carbapenemase recommendation in *P. aeruginosa* (35.32, 95% CI, 8.99–42.06).

### 3.3. Clinical Isolates with At Least One Carbapenem Intermediate Result

Isolates with non-susceptibility to any of the carbapenems evaluated were included. Thus, intermediate results were also included. Regarding *E. coli,* four clinical isolates had a value of 1 mg/mL for ertapenem (≤0.25 mg/mL for imipenem and ≤0.5 mg/mL for meropenem; Appendix A and Table 5). No carbapenemase result was detected in the Phoenix panels, and no carbapenemase-encoding gene was detected. All *K. pneumoniae* clinical isolates we included were resistant to at least one carbapenem (Appendix A).

Regarding *P. aeruginosa*, three strains were detected, with only one intermediate result (two for imipenem and two for meropenem). No carbapenemase recommendation was detected in two strains, and one strain had a recommendation of class D carbapenemase (imipenem 2 mg/mL and meropenem 4 mg/mL; Appendix A and Table 5).

## 4. Discussion

In this study, we used NMIC-501 CPO panels for the detection and classification of carbapenemases on a representative molecularly characterized collection of carbapenem non-susceptible *E. coli*, *K. pneumoniae*, and *P. aeruginosa* clinical isolates from Mexico. For *E. coli*, we detected a sensitivity of 70%, specificity of 83.33%, PPV of 94.38, and NPV of 40.98, with a prevalence of 80.00 for the detection of class B carbapenemases (*bla*_NDM_). This class of carbapenemases is currently the most frequently reported in Mexico for *E. coli* in several studies [22,23]. A study reported that high-prevalence countries may exhibit a low NPV, as was found in the present study (40.98%); thus, additional tests are required to exclude the presence of a CPO with a high probability [14].

Commercially available tests should be evaluated in different populations, depending on the prevalence of carbapenemases. For example, the most frequently reported carbapenemase-encoding gene in Mexico in recent years is *bla*_NDM_ [22,23]. In contrast, some countries, including the USA, Brazil, Colombia, and Argentina, have a high proportion of *bla*_KPC_ in Enterobacterales [24,25]. Thus, a different diagnostic utility is expected in these countries depending on the prevalence of circulating carbapenemases.

For *K. pneumoniae,* a sensitivity of 75%, specificity of 100%, PPV of 100.00%, and NPV of 95.59 with a prevalence of 15.58% were detected for class A carbapenemases (*bla*_KPC_ according to the PCR assay results). The high specificity value of this assay makes it reliable for detecting class A carbapenemases without the need to perform additional tests when a positive class A result is detected using the CPO panel. In contrast, if a negative result is detected, an additional confirmatory method must be performed because the presence of any carbapenemase cannot be discarded. The use of CPO panels may be useful especially if the test is positive.

*P. aeruginosa* may exhibit intrinsic or acquired resistance against nearly all available antibiotics. Multiple mechanisms may contribute to its antimicrobial resistance, including the production of carbapenemase enzymes, target mutations, the loss of outer membrane proteins, and multidrug efflux systems [26]. According to our CPO panel results, we calculated the performance of no detection of carbapenemase, for which a low sensitivity was detected (23.35%), which was associated with rather a high prevalence (76.61%). Thus, the CPO panels seem to have low utility for *P. aeruginosa*.

The use of this panel may reduce the time to obtain a result compared with the conventional approach because the result of the presence of carbapenemase (or not) is available as soon as the AST results are generated [14]. Overall, the detection of infection via a microorganism producing KPC-type β-lactamases is relevant because these enzymes are inhibited by the newer β-lactamase inhibitors, such as avibactam, relebactam, and vaborbactam, and not inhibited by clavulanate or tazobactam [27,28,29]. Thus, adequate treatment may be administered once the results are ready.

It is expected that the presence of more than one of the studied mechanisms and another molecular mechanism of antibiotic resistance, such as the presence of extended-spectrum β-lactamases (ESBL) or the presence of AmpCs, may have an impact on The specificity of Phoenix CPO panels [11]. In Mexico, the production of ESBL was reported to be as high as 39.3 in *K. pneumoniae* and 44.9 in *E. coli* [22], and some clinical isolates analyzed in this study were ESBL producers (Appendix A). However, the proportion was low, and we did not search for AmpCs with porin loss; thus, the impact of the presence of this gene cannot be measured.

Notably, the results observed in the CPO panels do not remove the need for additional laboratory tests to provide an informative antibiogram. For example, for *E. coli*, we found that 55 clinical isolates had no recommendation in the CPO report, and no carbapenemase-encoding gene was detected for 16 of them, (all these strains were non-susceptible to carbapenems), while at least one carbapenemase-encoding gene was detected for all other isolates. For *K. pneumoniae*, 60 isolates had no recommendation in the CPO report. No carbapenemase-encoding gene was detected for 11 of them, while all others detected at least one carbapenemase-encoding gene. Finally, for *P. aeruginosa*, 52 isolates had no recommendation in the CPO report. No carbapenemase-encoding gene was detected for 39 of them, while in the other 13 strains, *bla*_VIM_, *bla*_IMP_, and *bla*_GES_ were detected. The results may differ according to the prevalence of the carbapenemase-encoding gene, particular gene, and bacterial species [7,30,31].

In this study, the best positive likelihood ratio was 4.2 for *E. coli* for the detection of a class B carbapenemase (negative likelihood ratio, 0.36). With this value, the possibility of observing a positive result in clinical isolates with the presence of the enzyme versus the possibility of that positive result in strains negative for the enzyme is regular. The use of LR is a helpful tool because they are values inherent to the test and are not dependent on the prevalence of the disease. Thus, it may reflect the utility of the test in other populations.

In our study, we included clinical isolates collected from clinical samples between January 2021 and January 2023 from 12 centers in Mexico. The species included *E. coli*, *K. pneumoniae*, and *P. aeruginosa* with at least one result of intermediate to any of the carbapenem evaluated. We detected four clinical isolates for *E. coli* and three strains for *P. aeruginosa* (all *K. pneumoniae* clinical isolates were resistant to at least one of the carbapenems evaluated). The inclusion of clinical isolates with only one intermediate result may have decreased the diagnostic utility of the test; however, we decided to include it because some enzymes, especially those included in OXA (OXA-48, OXA-23, and OXA-24), may have a low carbapenemase activity and can go unnoticed [3].

The results of this study showed a limited utility for *P. aeruginosa*. In this bacterial species, the results for no carbapenemase recommendation had a sensitivity of 35.32%. It has been reported that in *P. aeruginosa*, carbapenem resistance may be associated in a significant proportion of strains with increased expression of efflux systems, reduced porin expression, and increased chromosomal cephalosporinase activity. Furthermore, other strains may produce carbapenemases plasmid or integron-mediated resistance [32]. In this study, PCRs were performed to determine if the carbapenemase-encoding gene was present or not; however, the presence of the gene does not ensure the production of a functional enzyme.

A limitation of this study was that our results represent the utility value for a country like Mexico, and the application of these panels depends on the bacterial species and the carbapenemase detected. Thus, this test should be interpreted with a deep knowledge of local epidemiology. Furthermore, the results observed in the CPO panels do not remove the need for additional laboratory tests to obtain an informative antibiogram.

## 5. Conclusions

In conclusion, Phoenix NMIC-501 panels may provide the results of the Ambler classification at the same time as the AST results. The CPO test is reliable for detecting class B carbapenemases in *E. coli,* but the relatively low specificity requires additional confirmatory methods. The carbapenemase classification in *K. pneumoniae* for class A carbapenemases has a high specificity and PPV; thus, a positive result is of high value. The CPO test has a limited utility for carbapenem resistant *P. aeruginosa.*

## Figures and Tables

**Figure 1 diagnostics-13-03417-f001:**
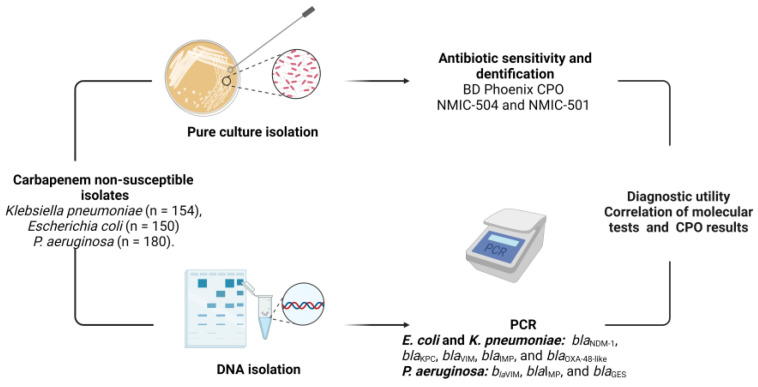
Methodology for determination of diagnostic utility of CPO panels. Adapted from “The 1-2-3 form of Health Care Associated Infections (HAI) research” from BioRender.com (2023). Retrieved from https://app.biorender.com/biorender-templates (accessed on 10 October 2023).

**Table 1 diagnostics-13-03417-t001:** Distribution of carbapenemase-encoding genes for *E. coli* detected according to the CPO carbapenemase report.

N	CPO Report	*bla* _NDM_	*bla* _KPC_	*bla* _VIM_	*bla* _IMP_	*bla* _GES_	*bla* _OXA48_
1	Class A	+	+	−	−		−
49	Class B	+	−	−	−		−
25	Class B	+	−	−	−		+
4	Class B	+	−	+	−		+
4	Class B	+	−	+	−		−
4	Class B	−	−	−	−		−
1	Class B	+	−	−	+		+
1	Class B	−	+	−	−		+
1	Class B	−	−	+	−		+
2	Class D	−	−	−	−		−
1	Class D	−	−	−	−		+
1	Carbapenemase producer	−	−	−	−		+
1	Carbapenemase producer	+	−	−	−		+
17	No recommendation	+	−	−	−		+
16	No recommendation	+	−	−	−		−
16	No recommendation	−	−	−	−		−
5	No recommendation	−	−	−	−		+
1	No recommendation	+	−	+	−		+

+: Positive, −: Negative.

**Table 2 diagnostics-13-03417-t002:** Distribution of carbapenemase-encoding genes for *K. pneumoniae* detected according to the CPO carbapenemase report.

N	CPO Report	*bla* _NDM_	*bla* _KPC_	*bla* _VIM_	*bla* _IMP_	*bla* _GES_	*bla* _OXA48_
11	Class A	−	+	−	−		−
3	Class A	+	+	−	−		+
3	Class A	−	+	−	−		+
1	Class A	−	+	−	+		+
33	Class B	+	−	−	−		−
29	Class B	+	−	−	−		+
5	Class B	−	−	−	−		−
2	Class B	+	−	+	−		+
1	Class B	−	+	−	−		−
1	Class B	+	−	+	−		−
1	Class B	−	−	+	−		−
1	Class D	−	−	−	−		+
1	Class D	−	−	−	−		−
1	Carbapenemase producer	−	−	−	−		−
1	Carbapenemase producer	+	−	+	−		−
20	No recommendation	+	−	−	−		+
18	No recommendation	+	−	−	−		−
11	No recommendation	−	−	−	−		−
3-1	No recommendation	−	−	+	−		+
2	No recommendation	+	−	+	−		+
2	No recommendation	−	+	−	−		+
1	No recommendation	−	+	−	−		−
1	No recommendation	−	+	+	−		+
1	No recommendation	−	+	+	−		−
1	No recommendation	−	−	−	+		−

+: Positive, −: Negative.

**Table 3 diagnostics-13-03417-t003:** Distribution of carbapenemase-encoding genes for *P. aeruginosa* detected according to the CPO carbapenemase report.

N	CPO Report	*bla* _NDM_	*bla* _KPC_	*bla* _VIM_	*bla* _IMP_	*bla* _GES_	*bla* _OXA48_
13	Class A			−	−	+	
6	Class A			−	−	−	
2	Class A			−	+	+	
1	Class A			+	−	−	
27	Class B			+	−	−	
6	Class B			−	−	−	
2	Class B			−	+	+	
1	Class B			+	+	−	
2	Class B			−	+	−	
20	Class D			−	−	−	
5	Class D			−	−	+	
1	Class D			+	−	−	
18	Carbapenemase producer			−	−	+	
6	Carbapenemase producer			−	−	−	
9	Carbapenemase producer			−	+	−	
3	Carbapenemase producer			−	+	+	
4	Carbapenemase producer			+	−	−	
2	Carbapenemase producer			+	−	+	
39	No recommendation			−	−	−	
4	No recommendation			+	−	−	
6	No recommendation			−	−	+	
3	No recommendation			−	+	−	

+: Positive, −: Negative.

**Table 4 diagnostics-13-03417-t004:** Performance of CPO-501 panels in detecting carbapenemase producer organisms.

	*E. coli,* Class B Value (95% CI)	*K. pneumoniae,* Class A Value (95% CI)	*E. coli* and *K. pneumoniae,* Class B Value (95% CI)	*P. aeruginosa*, No Recommendation Value (95% CI)
Sensitivity	70.00 (60.96–78.02)	75.00 (53.29–90.23)	63.14 (56.63–69.30)	23.35 (17.16–30.51)
Specificity	83.33 (65.28–94.36)	100.00 (97.20–100.00)	82.35 (71.20–90.53)	74.51 (60.37–85.67)
PLR	4.20 (1.87–9.43)	NA	3.58 (2.12–6.03)	0.92 (0.53–1.58)
NLR	0.36 (0.26–0.49)	0.25 (0.13–0.50)	0.45 (0.37–0.55)	1.03 (0.86–1.23)
Prevalence	80.00 (72.70–86.08)	15.58 (10.25–22.30)	77.63 (72.52–82.19)	76.61 (70.41–82.06)
PPV	94.38 (88.21–97.42)	100.00 (0.00–0.00)	92.55 (88.04–95.44)	75.00 (63.53–83.79)
NPV	40.98 (33.60–48.80)	95.59 (91.55–97.74)	39.16 (34.51–44.01)	22.89 (19.85–26.24)
Accuracy	72.67 (64.80–79.62)	96.10 (91.71–98.56)	67.43 (61.85–72.67)	35.32 (28.99–42.06)

PLR: positive likelihood ratio, NLR: negative likelihood ratio, PPV: positive predictive value, NPV: negative predictive value, CI: confidence interval, NA: not apply.

**Table 5 diagnostics-13-03417-t005:** *E. coli* and *P. aeruginosa* isolates with at least one carbapenem intermediate result.

ID	CPO Report	*bla* _NDM_	*bla* _KPC_	*bla* _VIM_	*bla*I_MP_	*bla* _OXA-48_	*bla* _GES_	ETP	IPM	MEM
*E. coli*										
21-0100	NR	-	-	-	-	-	ND	1	≤0.25	≤0.5
22-1191	NR	-	-	-	-	-	ND	1	≤0.25	≤0.5
22-1782	NR	-	-	-	-	-	ND	1	≤0.25	≤0.5
22-2012	NR	-	-	-	-	-	ND	1	1	≤0.5
*P. aeruginosa*										
21-700	NR	ND	ND	-	-	ND	-	ND	4	≤0.5
21-0280	NR	ND	ND	-	-	ND	-	ND	4	4
21-0788	Class D	ND	ND	-	-	ND	-	ND	2	4

ND: Not determined—Negative result. ETP: ertapenem, IPM: imipenem, MEM: meropenem. CPO: carbapenem producer organism.

## Data Availability

All information is included in the manuscript.

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
