# Peer review of "Evaluation of the BD Phoenix Carbapenemase-Producing Organism Panels for the Detection of Carbapenemase Producers in Escherichia coli, Klebsiella pneumoniae and Pseudomonas aeruginosa"

_diagnostics, 2023, doi:10.3390/diagnostics13223417_

Round 1
Reviewer 1 Report
Comments and Suggestions for Authors
The authors aimed to evaluate the BD Ohienix CPO panels for the detection of carbapenemase producers in E.coli, K. pneumoniae and P. aeruginosa and they found limited reliability for the carbapenemase detection.
They used the appropriate metholology.
Some comments
1)line 66-67 please check soundness and provide a reference
2)Please clearly state in the discussion and conclusion
i)for which carbapenemases and species the panel was found not to be reliable
ii) what is the added value if any of the test in the routine diagnostics
Comments on the Quality of English Language
Minor english editing is required
Author Response
Revisor 1
We appreciate the review of our manuscript. Please find below the response to each comment.
Some comments
1)line 66-67 please check soundness and provide a reference
Response: The sentence was changed to
Among class A β-lactamases, Klebsiella pneumoniae carbapenemase (KPC) hydrolyzes penicillins, cephalosporins, and carbapenems and is inhibited by clavulanate or tazobactam [3]. All text is in yellow for easy location.
2)Please clearly state in the discussion and conclusion
i)for which carbapenemases and species the panel was found not to be reliable
- ii) what is the added value if any of the tests in the routine diagnostics
Response:
The next paragraph, with the corresponding reference was added at the end of the discussion section:
The results of this study showed a limited utility for P. aeruginosa. In this bacterial species, the results for no carbapenemase recommendation have a sensitivity of 35.32 %. It has been reported that in P. aeruginosa, carbapenem resistance may be associated in a significant proportion of strains with increased expression of efflux systems, reduced porin expression, and increased chromosomal cephalosporinase activity. Furthermore, other strains may produce carbapenemases plasmid or integron-mediated.
Also, the next sentence was added to the conclusions:
The CPO test has a limited utility for carbapenem resistant P. aeruginosa.
Minor English editing is required
Response: English grammar was carefully reviewed
Reviewer 2 Report
Comments and Suggestions for Authors
The paper deals to evaluate the BD Phoenix CPO panels for the detection of carbapenemase producers among some bacteria.
The idea is publication-worth, the paper is well written, nevertheless, some issues should be adressed before consideration for publication and the English must be checked throughout the manuscript.
While the investigation deals the test of established commercial kit, it would be worth to explain the principle of carbapenemases detection, since it would help to unstand limitations of the assay. As well, this is not clear what has been detected - enzymatic acticity, presence of the gene (DNA PCR) or the gene expression (RT-PCR)? In many cases even the presence of the gene does not provide the presence of functional product.
section 2.3. Why 2 colonies were tested? in one assay or two independent assays? Please provide oligonucleotides used in the study. Also provide what was used as a positive control for PCR.
Author Response
We appreciate the review of our manuscript. Please find below the response to each comment.
Revisor 2
English must be checked throughout the manuscript.
Response: English was carefully reviewed
While the investigation deals the test of established commercial kit, it would be worth to explain the principle of carbapenemases detection, since it would help to understand limitations of the assay.
Response: The next sentence and the corresponding reference was added to the new manuscript:
The CPO panels include several carbapenems and other antibiotics previously reported to have activity on some Ambler classes including temocillin and cloxacillin. Furthermore, these panels include some β-lactamase inhibitors to characterize the carbapenemases according to Ambles class.
As well, this is not clear what has been detected - enzymatic activity, the presence of the gene (DNA PCR) or the gene expression (RT-PCR)? In many cases, even the presence of the gene does not provide the presence of functional product.
Response: The next sentence was added to the discussion section:
In this study, PCRs were performed to determine if the carbapenemase encoding gene were present or not; However, the presence of the gene does not ensure the production of a functional enzyme.
section 2.3. Why 2 colonies were tested? in one assay or two independent assays? Please provide the oligonucleotides used in the study. Also provide what was used as a positive control for PCR.
Response: We included 2 colonies, to collect a good amount of DNA for PCR. The two colonies were mixed and only one PCR was performed.
Oligonucleotides used in this study are described in the new Supplementary Table 1. All other supplementary tables were renamed.
For positive controls, we used strains ATCC BAA2468, ATCC BAA1905, and clinical isolates previously characterized P104, LMM-1873, LMM 1105, LMM 2935.